# The Advancement of 7XXX Series Aluminum Alloys for Aircraft Structures: A Review

**Bo Zhou, Bo Liu *** and **Shengen Zhang ***

Institute for Advanced Materials and Technology, University of Science and Technology Beijing, Beijing 100083, China; b20170572@xs.ustb.edu.cn
* Correspondence: liubo@ustb.edu.cn (B.L.); zhangshengen@mater.ustb.edu.cn (S.Z.)

**Abstract:** 7XXX series aluminum alloys (Al 7XXX alloys) are widely used in bearing components, such as aircraft frame, spars and stringers, for their high specific strength, high specific stiffness, high toughness, excellent processing, and welding performance. Therefore, Al 7XXX alloys are the most important structural materials in aviation. In this present review, the development tendency and the main applications of Al 7XXX alloys for aircraft structures are introduced, and the existing problems are simply discussed. Also, the heat treatment processes for improving the properties are compared and analyzed. It is the most important measures that optimizing alloy composition and improving heat treatment process are to enhance the comprehensive properties of Al 7XXX alloys. Among the method, solid solution, quenching, and aging of Al 7XXX alloys are the most significant. We introduce the effects of the three methods on the properties, and forecast the development direction of the properties, compositions, and heat treatments and the solution to the corrosion prediction problem for the next generation of Al 7XXX alloys for aircraft structures. The next generation of Al 7XXX alloys should be higher strength, higher toughness, higher damage tolerance, higher hardenability, and better corrosion resistance. It is urgent requirements to develop or invent new heat treatment regime. We should construct a novel corrosion prediction model for Al 7XXX alloys via confirming the surface corrosion environments and selecting the accurate and reliable electrochemical measurements.

**Keywords:** Al 7XXX alloys; machining technology; mechanical properties; corrosion properties; heat treatment; corrosion prediction

## 1. Introduction

With the development of modern technology, all walks of life have set off a wave of lightweight materials [1], especially in the aerospace and automotive fields. Many countries and enterprises are committed to in-depth research on new high-strength aluminum alloys and expect to reduce the weight of the materials to the maximum while maintaining the stability of mechanics and corrosion resistance for the overall structure, so as to replace traditional materials such as steel [2–5].

Influenced by COVID-19 in 2020, the global air passenger traffic has reduced, but the average annual growth of global air passenger traffic will reach 4.0% in 20 years based on Boeing's Report. 22,420 additional passenger and cargo aircrafts will be increased, while the total number of aircraft is expected to come up to more than double between 2019 and 2039 according to Boeing's forecast, which is demonstrated in Figure 1. In addition, approximately 20,690 low-fuel-efficient passenger and cargo aircrafts will be replaced by new [6,7]. The rapid development of the aviation industry contributes to the progress of new materials. Although the proportion of titanium alloys and composite materials has increased in newly developed aircraft, the use of high-strength aluminum alloys still accounts for a large proportion, which makes it indispensable in the aerospace field [8].

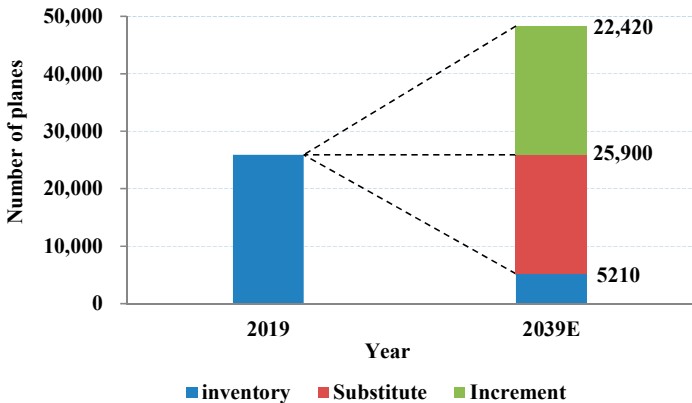

**Figure 1.** Boeing forecasts global aircraft demand [6].

As shown in Figure 2a, aircraft structural materials have shown a steady development trend from the Boeing 747 to the new generation of aircraft represented by the Boeing 777 and Airbus A380, and the amount of aluminum alloys used in civil aircraft accounts for more than 70%. In military aircraft, although the main structure materials have undergone great changes, it is still aluminum alloys that occupy the main position. As shown in Figure 2b, the proportion of aluminum alloys used in military aircraft is more than 35% except for F-22 [9]. According to statistics, global aviation aluminum demand has exceeded 2 million tons, with an annual average of more than 400 thousand tons from 2016 to 2020.

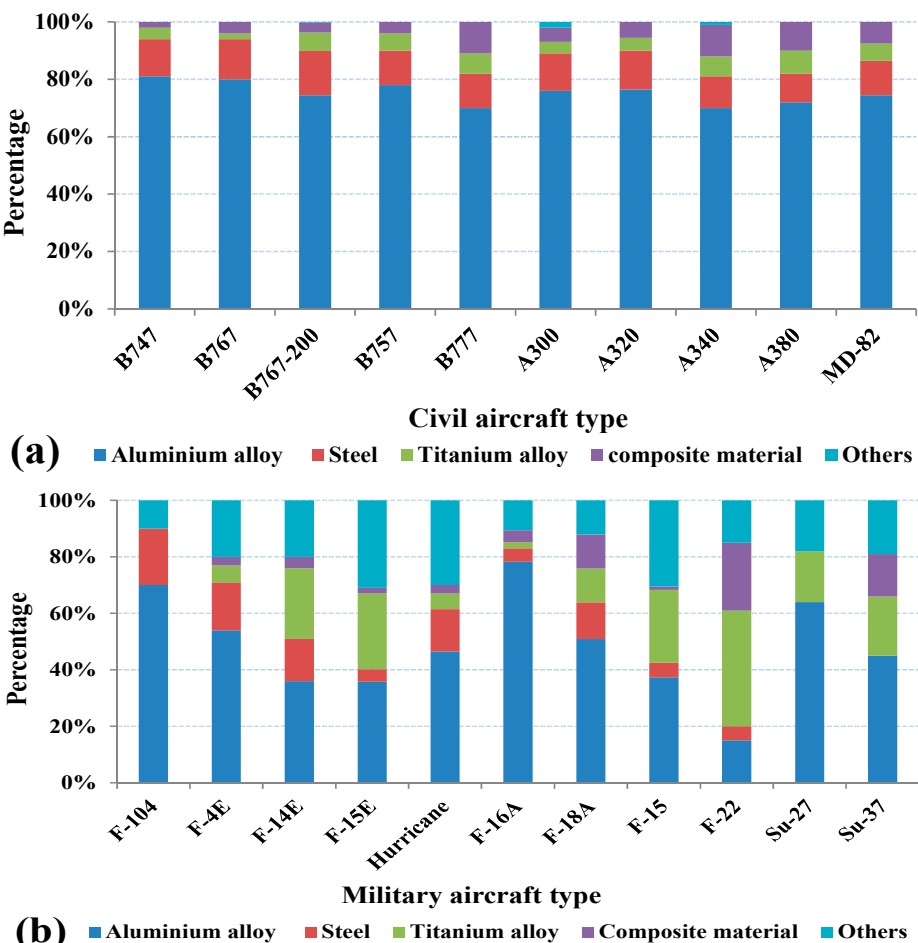

**Figure 2.** Ratio of materials used in Aircrafts: (**a**) civil (**b**) military [9].

Traditional aluminum alloys for aircraft structures are principally high-strength Al 2XXX alloys (2024, 2224, 2324, 2424, 2524, etc.) and ultra-high-strength Al 7XXX alloys (7075, 7475, 7050, 7150, 7055, 7085, etc.) [10]. After the 1980s, new materials including aluminum–lithium alloys, rapid solidification aluminum alloys, and aluminum matrix composite materials have been rapidly developed [11,12]. The traditional 2XXX series and Al 7XXX alloy materials were greatly affected, but they still show strong vitality with their superior performance [13–16]. According to statistics, high-performance 2XXX series and Al 7XXX alloys are still mostly used in more than 70% structural materials of aircraft. According to the application classification, aviation aluminum can be divided into casting aluminum alloy, extruded aluminum alloy, rolled aluminum alloy, forged aluminum alloy, etc. [17] The aluminum proportional distribution for aviation is shown in Figure 3.

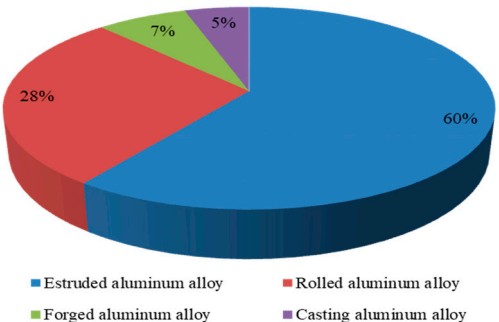

**Figure 3.** Aluminum proportional distribution for aviation (mass fraction, %) [9].

As the main part of high-strength and high-toughness aluminum alloy, Al 7XXX alloys have high specific strength, high specific stiffness, high toughness, excellent processing and welding performance. They are widely applied in the manufacture of aircraft frame, spars, and stringers as load-bearing components, and have become one of the most important structural materials in this field [18].

Due to the combined effect of the service environment and the bearing load during actual use, the stress corrosion has always been a fatal defect of Al 7XXX alloys as aircraft structural materials, which has caused many aircraft accidents [19]. Therefore, in order to deal with the problems in the harsh and long-term working environment during the service process of the aircraft, the aircraft structural materials not only need high static strength, high fracture toughness, high fatigue strength, and excellent high temperature performance, it is also necessary to have good stress corrosion resistance. As a response to these various needs, the heat treatment processes of Al 7XXX alloys are correspondingly produced, mainly including homogenization treatment, thermal deformation like rolling, extrusion, solid solution, quenching, and aging treatment [20,21].

This paper reviews the development tendency, requirement stage, main bands and applications, joining and milling techniques, and additive manufacturing technology of Al 7XXX alloys for aircraft structures, and the main problems faced in the applications are pointed out. Furthermore, the main measures to improve the performance are discussed, mainly introducing the effects of solution, quenching and aging processes on the properties. Moreover, the development direction to the new generation of Al 7XXX alloys for aircraft structure and the corrosion prediction are discussed.

## 2. Development of Al 7XXX Alloys for Aircraft Structures

### 2.1. Development Tendency

From 1923 to 1924, German scientists Sander and Meissner et al. [22] discovered that the Al-Mg-Zn alloy formed $MgZn_2$ strengthening phase after solid solution, quenching and aging treatment, which greatly improved the strength. Since then, research on Al 7XXX alloys has begun. In 1932, Webber et al. found that a small amount of Cu and Mn elements

contained in Al-Mg-Zn alloys can improve their stress corrosion resistance, and developed the earliest Al 7XXX alloys, which later Al 7XXX alloys are based on to develop.

From 1935 to 1939, Japanese scientist Igarashi added Cr, Mn, and Mo to Al-Mg-Zn-Cu alloys to develop a new high-strength extra super duralumin alloy, which was first used in carrier-based aircrafts because of its high strength and good stress corrosion resistance. Soon afterwards, the United States developed the Al 7075 alloy in 1943 and applied it to the B-29 bomber, which greatly promoted the development of high-strength aluminum alloys in the aviation field. In 1948, the former Soviet Union developed B95 aluminum alloy which was similar to the Al 7075 alloy.

In 1954, the United States developed an Al 7178-T651 alloy with a higher strength than Al 7075-T651 alloy by increasing the alloy element content on the basis of Al 7075 alloy, which was used on Boeing 707, 737, and DC8 passenger aircraft. In 1956, the former Soviet Union added Zr element to Al-Zn-Mg-Cu series alloys for the first time to suppress the recrystallization, which developed B96u aluminum alloy with high strength and high alloy degree [23]. In 1960, the double aging process T73 was developed and applied to Al 7075 alloy to reduce the susceptibility of stress corrosion and exfoliation corrosion, which especially solved the problem of stress corrosion on thick sections. In the mid-1960s, the T76 aging process was developed, which improved the strength of the alloy compared to the T73 state and met the requirements for the resistance to stress corrosion and exfoliation corrosion [24,25].

In 1968, based on the Al 7001 alloy, the content of Cu and Cr was reduced, and the Zn/Mg ratio was increased, while the toughness and stress corrosion resistance of the alloy were improved, so that the Al 7049 alloy was successfully developed. In 1969, the Al 7475 alloy with the highest fracture toughness among Al-Zn-Mg-Cu series alloys was successfully developed, which was made on the basis of the improved purity of the Al 7075 alloy [26].

In 1971, based on the Al 7075 alloy, the United States increased the Zn and Cu content, and the Cu / Mg ratio to increase the strength of the alloy. They also added Zr instead of Cr to overcome the quenching sensitivity and adjust the grain size, which means Al 7050 alloy was exploited with higher strength, fracture toughness, and stress corrosion resistance. In 1978, based on the optimization of the main Al 7050 alloy composition, American Alcoa increased the Zn content and reduced the quantity of the Fe and Si impurity phases, thus developed a new Al 7150 alloy with better toughness and resistance to exfoliation corrosion.

In 1989, the Alcoa developed the T77 treatment process and applied to 7150 alloy to obtain the T6 state strength and T73 state corrosion resistance, and the strength of the Al 7055 alloy through T77 aging was higher. In 1992, Japanese Sumitomo Light Metals Co., Ltd. produced ultra-high-strength aluminum alloys with the tensile strength of more than 700 MPa in the laboratory using vacuum compacting and sintering processes. At the end of the 1990s, the United States, Japan, and other nations developed a new generation of ultra-high-strength aluminum alloys with the Zn content of more than 8 wt %, the tensile strength of 760-810 MPa, and the elongation of 8–13% by using spray forming technology. It was used for manufacturing structural parts in transportation and other high-stress structural parts that required high strength and resistance to stress corrosion.

In 2003, Alcoa launched a new generation of high-strength and high-toughness Al 7085 alloy. Due to its good casting properties and excellent hardenability, it has great application potential in new-generation aircraft components. It has been reported that the comprehensive properties of Al 7085 alloy has exceeded Al 7050 alloy. At the same process, the stress corrosion resistance and fracture toughness of Al 7085 alloy component are equivalent to Al 7050 alloy, but its strength can be increased by 15%, and its maximum thickness is up to 305 mm. Until now, Al 7085 alloy is one of the most advanced aluminum alloys in the world [3,18,25,27–29].

In short, the development law of Al 7XXX alloys is based on the existing properties to greatly improve other aspects of the alloy. Specifically, in terms of alloy design, the content of impurities such as Fe and Si is getting lower, and the alloying degree is getting

higher, while the trace transition group elements are becoming more reasonable, as shown in Figure 4.

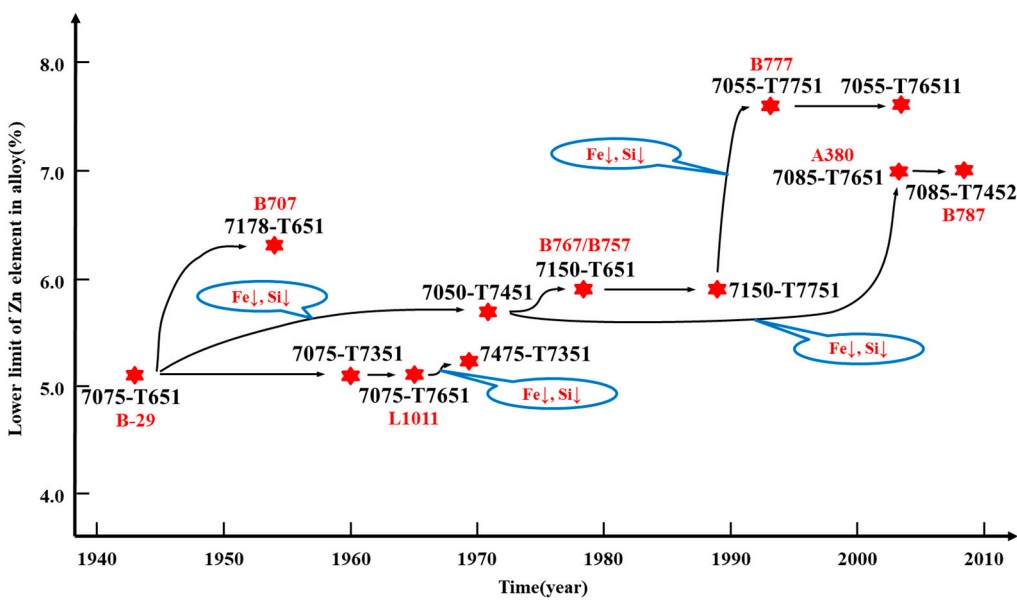

**Figure 4.** Flow chart of the development of Al 7XXX alloys [3,18,22–29].

*2.2. Requirement Stages*

As shown in Figure 5, the development of Al 7XXX alloys could be roughly divided into five stages [2,13,26].

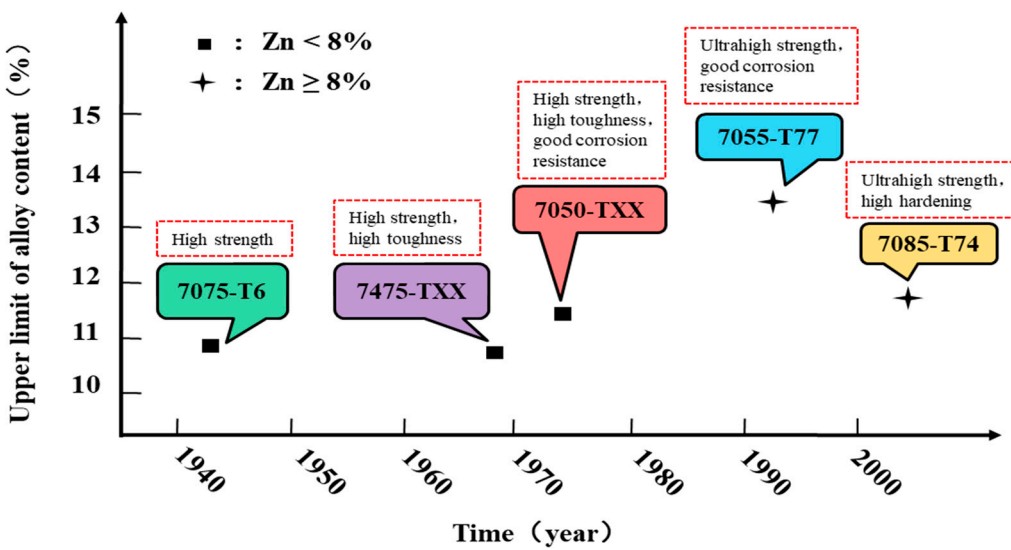

**Figure 5.** The development process of Al 7XXX alloys [2,13,26,30].

The first generation of Al 7XXX alloys is represented by high-strength Al 7075-T6 alloy. At this stage, the purpose is to improve the static strength. Nevertheless, the importance of fracture toughness and corrosion resistance has not been taken into account. The representative of the second generation is Al 7475-TXX alloy. At this stage, the corrosion resistance of the alloy is improved at the expense of reducing the strength. The representative of the third generation is Al 7050-TXX alloy. At this stage, the comprehensive properties of the alloy are pursued, especially in strength, fracture toughness and stress corrosion resistance. The representative of the fourth generation is Al 7055-T77 alloy. At this stage,

the contradiction between strength and stress corrosion resistance was ameliorated. For instance, Alcoa developed Al 7055-T77 alloy with higher strength and better resistance corrosion performance based on retrogression and re-aging (RRA). The representative of the fifth generation is Al 7085-T74 alloy. At this stage, it aims to develop the aluminum alloys with ultra-high strength, high toughness, and high hardenability [30]. Through adjusting the composition of the alloy, the Al 7085 alloy was born to satisfy the needs of strength, quench sensitivity, fatigue performance, and stress corrosion resistance to meet the development of the aviation industry and the urgent needs of the large aircraft industry.

### 2.3. Main Brands and Applications

Al 7XXX alloys can be strengthened by heat treatment, which mainly contained Zn element. The Al-Zn-Mg alloy, which means that Mg is added to the alloy, is a high-strength and weldable aluminum alloy and has good thermal deformation properties and a wide quenching range. On appropriate conditions of heat treatment, it can obtain higher strength, better welding performance, and better resistance to corrosion performance [31,32]. Al-Zn-Mg-Cu alloy is developed by adding Cu on the basis of Al-Zn-Mg alloy. Its strength is higher than Al 2XXX alloys, which is generally called ultra-high-strength aluminum alloy. The yield strength of the alloy is close to the tensile strength. The specific strength is also very high, but the plasticity and high-temperature strength are low. It is suitable to be used as a load-bearing structural component at room temperature or below 120 °C. The alloy is easy to process, and has good corrosion resistance and high toughness [28,33]. The main brands and chemical compositions of Al 7XXX alloys are shown in Table 1.

**Table 1.** Main brands and chemical compositions of Al 7XXX alloys (mass fraction, %) [2,34–36]

| Alloy | Zn | Mg | Cu | Cr | Fe | Si | Mn | Ti | Zr | Other Each | Other Total | Al |
|---|---|---|---|---|---|---|---|---|---|---|---|---|
| 7075 | 5.1~6.1 | 2.1~2.9 | 1.2~2.0 | 0.18~0.28 | ≤0.50 | ≤0.40 | ≤0.30 | ≤0.20 | – | ≤0.05 | ≤0.15 | Bal. |
| 7178 | 6.3~7.3 | 2.4~3.1 | 1.6~2.4 | 0.18~0.28 | ≤0.50 | ≤0.40 | ≤0.30 | ≤0.20 | – | ≤0.05 | ≤0.15 | Bal. |
| 7001 | 6.8~8.0 | 2.6~3.4 | 1.6~2.6 | 0.18~0.35 | ≤0.40 | ≤0.35 | ≤0.20 | ≤0.20 | – | ≤0.05 | ≤0.15 | Bal. |
| 7079 | 3.8~4.8 | 2.9~3.7 | 0.40~0.80 | 0.10~0.25 | ≤0.40 | ≤0.35 | 0.10~0.25 | ≤0.10 | – | ≤0.05 | ≤0.15 | Bal. |
| 7175 | 5.1~6.1 | 2.1~2.9 | 1.2~2.0 | 0.18~0.28 | ≤0.20 | ≤0.15 | ≤0.10 | ≤0.10 | – | ≤0.05 | ≤0.15 | Bal. |
| 7179 | 3.8~4.8 | 2.9~3.7 | 0.40~0.80 | 0.10~0.25 | ≤0.20 | ≤0.15 | 0.10~0.30 | ≤0.10 | – | ≤0.05 | ≤0.15 | Bal. |
| 7049 | 7.2~8.2 | 2.0~2.9 | 1.2~1.9 | 0.10~0.22 | ≤0.35 | ≤0.25 | ≤0.20 | ≤0.10 | – | ≤0.05 | ≤0.15 | Bal. |
| 7475 | 5.2~6.2 | 1.9~2.6 | 1.2~1.9 | 0.18~0.25 | ≤0.12 | ≤0.10 | ≤0.06 | ≤0.06 | – | ≤0.05 | ≤0.15 | Bal. |
| 7050 | 5.7~6.7 | 1.9~2.6 | 2.0~2.6 | ≤0.04 | ≤0.15 | ≤0.12 | ≤0.10 | ≤0.06 | 0.08~0.15 | ≤0.05 | ≤0.15 | Bal. |
| 7049A | 7.2~8.4 | 2.1~3.1 | 1.2~1.9 | 0.05~0.25 | ≤0.50 | ≤0.40 | ≤0.50 | – | ≤0.25 | ≤0.05 | ≤0.15 | Bal. |
| 7009 | 5.5~6.5 | 2.1~2.9 | 0.60~1.3 | 0.10~0.25 | ≤0.20 | ≤0.20 | ≤0.10 | ≤0.10 | – | ≤0.05 | ≤0.15 | Bal. |
| 7109 | 5.8~6.5 | 2.2~2.7 | 0.8~1.3 | 0.04~0.08 | ≤0.15 | ≤0.10 | ≤0.10 | ≤0.10 | – | ≤0.05 | ≤0.15 | Bal. |
| 7010 | 5.7~6.8 | 2.1~2.6 | 1.5~2.0 | ≤0.05 | ≤0.15 | ≤0.12 | ≤0.10 | – | 0.11~0.17 | ≤0.05 | ≤0.15 | Bal. |
| 7012 | 5.8~6.5 | 1.8~2.2 | 0.8~1.2 | ≤0.04 | ≤0.25 | ≤0.15 | 0.08~0.15 | 0.04~0.08 | – | ≤0.05 | ≤0.15 | Bal. |
| 7149 | 7.2~8.2 | 2.0~2.9 | 1.2~1.9 | 0.10~0.22 | ≤0.20 | ≤0.15 | ≤0.20 | ≤0.10 | – | ≤0.05 | ≤0.15 | Bal. |
| 7150 | 5.9~6.9 | 2.0~2.7 | 1.9~2.5 | ≤0.04 | ≤0.15 | ≤0.15 | ≤0.10 | ≤0.06 | 0.08~0.15 | ≤0.05 | ≤0.15 | Bal. |
| 7278 | 6.6~7.4 | 2.5~3.2 | 1.6~2.2 | 0.17~0.25 | ≤0.20 | ≤0.15 | ≤0.02 | ≤0.03 | – | ≤0.05 | ≤0.15 | Bal. |
| 7055 | 7.6~8.4 | 1.8~2.3 | 2.0~2.6 | ≤0.04 | ≤0.10 | ≤0.10 | ≤0.05 | ≤0.06 | 0.05~0.25 | ≤0.05 | ≤0.15 | Bal. |
| 7249 | 7.5~8.2 | 2.0~2.4 | 1.3~1.9 | 0.12~0.18 | ≤0.12 | ≤0.10 | ≤0.10 | ≤0.06 | – | ≤0.05 | ≤0.15 | Bal. |
| 7085 | 7.0-8.0 | 1.2-1.8 | 1.3-2.0 | ≤0.04 | ≤0.08 | ≤0.06 | ≤0.04 | ≤0.06 | 0.08~0.15 | ≤0.05 | ≤0.15 | Bal. |

As the main part of high-strength and high-toughness aluminum alloy, Al 7XXX alloys have high specific strength, high specific stiffness, high toughness, excellent processing, and welding performance. They are widely applied in the manufacture of aircraft frame, spars, and stringers as load-bearing components, and have become one of the most important structural materials in this field [18,37]. The performance requirements of the aircraft materials are shown in Figure 6 [25,35,38].

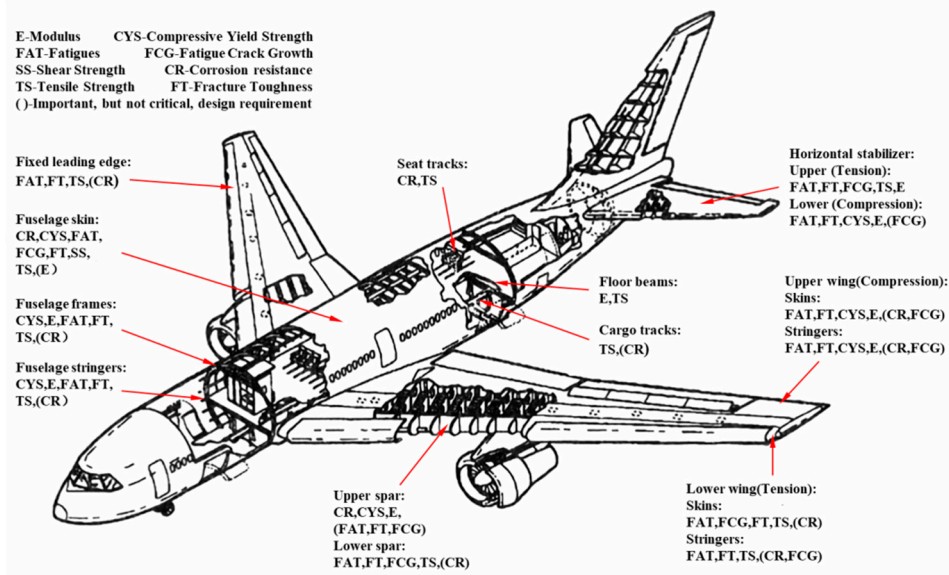

**Figure 6.** Performance requirements of the aircraft materials [25,35,38,39]. (Adapted with permission from ref. [35]. Copyright 1996 Elsevier).

The main features and applications of Al 7XXX alloys are shown in Table 2. It can be seen that Al 7075 alloys are mainly used for fuselage bulkheads, wing upper skins, wing upper panels, and vertical tails. Al 7475 alloy is mainly used for wing skins. Al 7050 alloys are mainly used in the fuselage bulkheads. Al 7055 alloys are mainly used for long fuselage stringers, fuselage beams, and fuselage upper stringers. Al 7085 alloys are mainly used for wing spars and ribs.

**Table 2.** Main features and applications of Al 7XXX alloys [28,40–42].

| Alloy | Main Features | Main Products and Status | Main Applications |
|---|---|---|---|
| 7075 | High strength at low temperature, poor weldability, poor SCC resistance | T6 T73 T76 sheet plate, T651 T7651 T7351 thick plate, T6 T73 T7352 casting, T6511 T3511 extrusion | Upper and lower wings skins, stringers, frames |
| 7049 | Instead of 7079, high strength, good SCC resistance | T3511 T76511 extrusion, T73 T7352 forgings, T73 sheet and thick plate | Main landing gear |
| 7149 | Good strength, good SCC resistance, better fracture toughness than 7049 | T73 T74 T7452 forgings, T7351 T76511 extrusion | Main landing gear |
| 7249 | Better comprehensive properties than 7149 | T73 T74 T7452 forgings, T7351 T76511 extrusion | Main landing gear |
| 7175 | High strength, good corrosion resistance, | T66 T74 T7452 forgings, T74 T6511 extrusion | Spars, main landing gear |
| 7475 | High strength, high fracture toughness, good fatigue resistance, good corrosion resistance | T61 T761 sheet plate, T651 T7351 sheet and thick plate | Fuselage skins, wing skins, central wing structures, spars, bulkheads |
| 7050 | High strength, good fracture toughness, good SCC resistance, good EXCO resistance, poor hardening sensitivity | T651 T7451 thick plate, T3511 T76511 T73511 T74511 extrusion, T7452 T76 T7652 T74 forgings, T73 wire, T76 sheet plate | Fuselage frame, wing skins, bulkheads, stringers, stiffener |
| 7150 | High strength, good corrosion resistance, good fatigue resistance | T651 T7751 thick plate, T6511 T77511 extrusion, T77 forgings | Upper wing structures, fixed leading edge, upper wing stringers, fuselage reinforcement, skeletons, seat tracks |
| 7055 | Higher compressive strength and tensile strength than 7150, similar fracture toughness and corrosion resistance to 7150 | T7751 thick plate, T77511 extrusion, T77 forgings | Upper wing skins, stringers, horizontal stabilizer, skeletons, seat tracks, cargo tracks |
| 7085 | Good comprehensive properties, high hardening, high strength, high fracture toughness, good SCC and EXCO resistance | T7651 thick plate, T7452 forgings | Wing spars and stringers for A380 |

### 2.4. Joining and Milling Techniques

A large number of high-precision connecting holes need to be processed to fasten the dissimilar stacks of the fuselage together with bolts and rivets in the aircraft assembly process. Whereas in fuselage manufacturing, the drilling operations of dissimilar stack materials are crucial [43]. Traditional drilling will cause several defects such as delamination, tear and burr, so that the machining accuracy is difficult to guarantee. Therefore, the drilling studies have been aroused by many scholars, mainly including drilling defect research, tool optimization, drilling parameter optimization, and vibration assisted drilling technology.

Denkena et al. [44] applied the helical milling technology to the drilling of CFRP–titanium layer compounds and studied the influence of different process parameters on the processing quality. Zitoune et al. [45] found that longer metal chips lead to a decrease in the surface quality of the composite material layer's pores, and good broken chips can improve hole quality and extend tool life. Wang et al. [46] studied the tool wear during drilling experiments on stack of carbon fiber reinforced plastic and TiAl6V4 and found that the different wear mechanisms of the two materials caused the drills to wear too quickly. Brehl and Dow et al. [47] studied the kinematic relationships of 1D (linear vibratory tool path) and 2D VAM (circular/elliptical tool path) vibration-assisted machining, and found that the intermittent contact between the tool and the uncut material workpiece reduces friction and heating, and improves surface quality and machining accuracy, so that extends tool life. Lacalle et al. [43] analyzed and modeled the chip formation process of drilling assisted by low-frequency vibrations of FC/Al stack material, and found that the problems for the final geometrical quality of the hole and burr formation can be avoided by the chip segmentation during the drilling operation resulting in less temperature increasing.

However, riveting increases the weight of the fuselage and also causes stress concentration which leads to fatigue crack initiation and growth. In order to reduce the weight and the inspection and maintenance costs for the aircrafts, new trends in the construction and manufacture of aircraft fuselage have therefore emerged in which friction-stir welding, laser beam welding, and milling machining are increasingly replacing the use of bolts and rivets [2,48].

In order to achieve the purpose of lightweight and meet the requirements of aircraft performance, many skeleton parts, especially main load-bearing structural parts, such as aircraft beams, bulkheads, and wall panels, are generally processed into complex grooves, ribs, bosses, lightening holes, and other integral structural parts that are directly hollowed out from a large block of blanks. Therefore, aviation integral structural parts have the characteristics of complex structure, large size, high material removal rate, and many thin-walled structural parts like frames, cantilever beams, and wall panels [49].

Due to poor rigidity, thin structural parts are affected by factors such as cutting force, cutting heat, cutting chatter, and part cutter relieving during milling. Therefore, they are prone to deformation, which seriously affects the dimensional accuracy and structure function.

The main forms of thin structural part deformation are machining vibration deformation, cutter relieving deformation, and overall machining deformation. Whereas the factors that affect the overall machining deformation are mainly the material properties of the workpiece, tongs layout design, process parameters, tool path strategies, etc.

In 1997, Tlusty et al. [50] proposed a tool path optimization scheme for thin-walled part deformation by effectively utilizing the unprocessed part of the components as support, so as to make full use of the integral rigidity of the components.

In 2004, Ratchev et al. [51] established the analytical flexible force model considering the geometric characteristics, the immersion angle and the material properties of the tool. Combined with the finite element technology, they put forward the static machining error compensation model for low rigidity components, providing further guidance for NC verification techniques.

In 2005, Herranz et al. [49] proposed an approach for the right selection of cutting conditions in high-speed milling of low-rigidity parts to avoid the static and dynamic problems, local and global structure deformations, and vibration. It has been applied in the actual production process, and the unrecovered parts have been reduced by 20–26% as calculated.

In 2008, Jitender et al. [52] developed a milling simulation system based on finite element. The system can predict the part thin wall deflections and elastic plastic deformations during machining by considering the effects of fixturing, operation sequence, tool path, and cutting parameters on transient thermal loading conditions. The numerical simulation results of cutting force and deformation obtained are in good agreement with the experimental data.

In 2016, Ismail et al. [53] established the functional relationship between the mechanical and thermal loads on the workpiece and the machining parameters to apply the combined effect to the thin part and proposed a new multi-physics based finite element modeling (FEM) approach to predict thin part deformation in micromilling. The simulation results are verified by real-time laser measurement and white light interferometer measurement with the average 14% error.

In 2019, Li et al. [54] constructed a semi-analytical model considering biaxial blank residual stress to predict machining deformations of five thin-walled parts with different stiffening rib layouts, and the accuracy of the model was verified by FEM simulations and machining experiments. The results show that the machining deformation decreases with the increase of equivalent bending stiffness in the length direction, and the equivalent stiffness in the width direction has little effect on the overall machining deformation.

## 2.5. Additive Manufacturing Technology

Due to the excellent strength-to-weight and stiffness-to-weight ratios with good machinability, Al 7XXX alloys are widely applied to manufacturing structural components in the aerospace industry. In traditional subtractive manufacturing processes, the geometrical complexity of many aerospace components bought many difficulties. However, additive manufacturing (AM) processes are widely used in this field because of the small size, high value, and geometrical complexity for the components during the manufacturing processes. Furthermore, through designing and manufacturing the components with complex topologies, AM processes reduce the total number of the aircraft parts to enable part consolidation [55]. The part consolidation brings many benefits including lower production cost, component failure risk, better product properties like high strength-to-weight ratio and lightweight, and lower material usage with part complexity increasing. Therefore, mass AMed components have been adopted to the aerospace industry [56].

AM processes, also known as 3D printing, produce the complex geometrical components layer by layer on the basis of three-dimensional (3D) data obtained by scanning physical objects or using design software. The representatives of AM technologies include selective laser sintering (SLS), selective laser melting (SLM), laser near net shaping (LENS), electron beam melting (EBM), wire arc additive manufacturing (WAAM) [57].

In 1995, the Fraunhofer Laser Technology Institute in Germany first carried out selective laser melting (SLM) forming technology research [58]. This technology directly melts metal powder by selecting appropriate process parameters to obtain components with high density. It shows that aluminum alloys are easy to oxidize, and have high reflectivity to lasers, so that SLM forming is more difficult.

In 2011, Bartkowiak et al. [59] took the lead in developing SLM forming research of high-strength aluminum alloy and research the SLM forming feasibility for Al-Cu and Al-Zn powder by low power fiber laser. Since then, the research on SLM forming of high-strength aluminum alloy has attracted the attention of the industry and developed rapidly in recent years.

At present, the research on SLM forming of high-strength aluminum alloys mainly focuses on Al-Cu alloys and Al-Zn-Mg-Cu alloys. Compared with Al-Si alloys, Al-Cu

alloys, and Al-Zn-Mg-Cu alloys are more difficult to form in SLM. Due to the wider solidification interval, greater hot cracking tendency, higher thermal conductivity, and higher alloy element content, higher laser energy is required during the forming process, and it is easy to cause element burning loss, so the additive manufacturing technology of high-strength aluminum alloys develops slowly [57].

In 2016, Kaufmann et al. [60] studied the influence of SLM process parameters on the forming quality of Al 7075 alloy, and finally obtained a sample with a density greater than 99% by optimizing the parameters. However, a preheating temperature of up to 200 °C did not show a significant positive effect on reduction of hot cracks.

In 2016, Sistiaga et al. [61] added 4 wt % silicon to Al 7075 alloy powder, which increased the density of SLM processed aluminum alloy to 99% and the hardness to 171 HV, reaching the conventional 7075 hardness level treated with T6. The study show that cracks can be avoided and mechanical properties can be improved by adding appropriate alloying elements.

In 2017, Singh et al. [62] added nickel to Al 7050 alloy powder sediments, and the brittle $Al_3Ni$ intermetallic was formed due to Ni segregation at the dendritic boundary, resulting in poor tensile ductility of Al 7050 alloy. Laser deposited samples by friction stir processed (FSP). The $Al_3Ni$ particles in α-Al matrix are refined and uniformly distributed. The yield strength, tensile strength, and elongation of the aluminum alloy are up to 178 MPa, 302 MPa and 6%, respectively. Moreover, after FSP heat treatment, the strength and elongation are increased by about 10%.

In 2017, Martin et al. [63] published a high-performance SLM forming method for 7XXX series ultra-high-strength aluminum alloys. The tensile strength, yield strength and elongation of the formed Al 7050 alloy samples after T6 heat treatment are 383–417 MPa, 325–373 MPa, and 3.8%–5.4%, respectively.

## 3. Main Problems of Al 7XXX Alloys for Aircraft Structures

### 3.1. Performance Problems

Due to the combined effect of the service environment and the bearing load during actual use, the stress corrosion has always been a fatal defect of Al 7XXX alloys as aircraft structural materials, which has caused many aircraft accidents, thus greatly limiting the applications [19,64]. According to the literature, the main stress corrosion cracking (SCC) mechanism in Al 7XXX alloys is anodic dissolution assisted cracking, hydrogen induced cracking, and passive film rupture, as shown in Figure 7 [65].

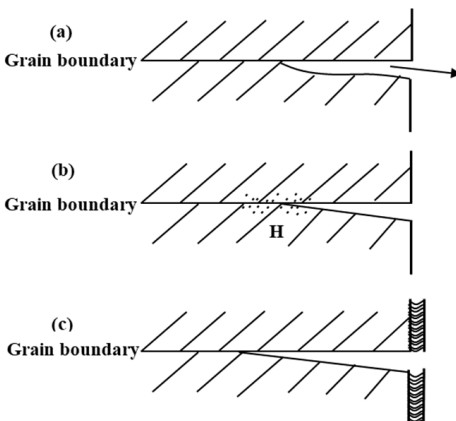

**Figure 7.** Illustration of mechanisms of SCC for aluminum alloys. (**a**) Anodic dissolution assisted cracking; (**b**) Hydrogen induced cracking; (**c**) Passive film rupture. (Reprinted with permission from ref. [65]. Copyright 2016 Elsevier).

Anodic dissolution assisted cracking mechanism is shown in Figure 7a. It is a typical intergranular corrosion failure mode. When the grain boundaries or grain adjacent regions are anodic against to the rest of the microstructure, the anodic dissolution can proceed selectively along the boundaries. Hydrogen induced cracking mechanism is shown in Figure 7b. The cathode reduction reaction occurs in the alloy to generate the hydrogen atoms, and some of the hydrogen atoms diffuse to the alloy interior. When the hydrogen atoms are supersaturated in the alloy, they will combine to form $H_2$ at microscopic defects. As $H_2$ concentration increases, the hydrogen pressure rises. When the stress generated by the hydrogen pressure is higher than the yield strength of the aluminum alloy, it will be generated partial plastic deformation so that the surface bulges to form hydrogen bubbles, which causes internal cracks. Passive film rupture mechanism is shown in Figure 7c. There is an oxide film on the surface of the aluminum alloy, which has a certain protective effect on the matrix. However, the oxide film can be damaged in $Cl^-$-rich atmospheric environment, causing localized corrosion processes like pitting. In addition, due to the high alloy element content of Al 7XXX alloys, the formed high-density precipitated phases are enriched in a chain-like manner at the grain boundaries, resulting in significant alloy cracking, which leads to stress corrosion and reduces the service life of the components [66–68].

### 3.2. Corrosion Prediction

The environmental research during the service of the aircraft found that while the structural parts of aircraft are in different positions, the corrosion environment will be different, so that the corrosion types and corrosion mechanisms will be different. Therefore, the corrosion behavior of Al 7XXX alloys for aircraft structures is not only a very complicated engineering phenomenon, but also a multidisciplinary scientific problem. It has become an urgent problem that how to construct a reasonable prediction model to accurately predict the corrosion behavior of Al 7XXX alloys for aircraft structures.

With the application of computers and the development of solution electrochemical measurement technology, a series of prediction models were developed to accurately predict the corrosion behavior of Al 7XXX alloys for aircraft structures. At present, the most common prediction models are based on corrosion electrochemical principle. In 1964, Fleck [69] used the finite difference method (FDM) for the first time to evaluate the current density distribution of the electrode system. At the same time, Klingert et al. [70] also explored the current density distribution of the electrode system via a high-speed computer. At the end of the 1970s, Alkire et al. [71] obtained the secondary potential field distribution in the electrolysis cell through finite element method (FEM) and predicted the change of electrode shape. Afterwards, corrosion prediction was applied to the field of cathodic protection for marine structures, and engineering applications were gradually realized. Although the FDM and FEM method gave relatively accurate results in many practical applications, some complex structures, and infinite domain problems could not be handled, due to the limitation of the calculation level at that time. Fu and Chow introduced the more efficient boundary element method (BEM) into the corrosive electric numerical calculation field first and proved the accuracy of this method. Helle et al. [72] used and compared two numerical methods when solving the galvanic corrosion problem of ships and propellers in seawater. Zamani et al. [73] completed a numerical simulation of a Canadian warship's cathodic protection system through the boundary element method. Comparison of numerical analysis methods are shown in Table 3.

**Table 3.** Comparison of numerical analysis methods [74–84]

| Prediction Models | Finite Element Method (FEM) | Finite Difference Method (FDM) | Boundary Element Method (BEM) |
|---|---|---|---|
| Precision of solution | Approximate solution, precision determined by grid partition | | |
| Applicable geometry | Solving complex geometry problems | | |
| Method of solving voltage | Determined by the data of a large number of nodes | | |
| Method of solving current density | Calculated by the voltage value with the same accuracy | | Calculated directly by the data at the boundary |
| Boundary conditions | Known potential at the boundary; Known current density at the boundary; known function of potential and current density at the boundary | | |
| Electrolyte characteristic | Uniform; Heterogeneous continuity; Bounded and conductive | | Uniform and continuously conductive |
| Solution domain | Finite field | | Infinite domain |
| Number of equations | The same number as nodes that are distributed on the entire domain | | The same number as nodes at the boundary |

The theory and technology for numerical simulation and prediction of corrosion are becoming more and more advanced with a large number of researchers. Related scientific research institutes have successively developed a series of corrosion protection prediction and design software, such as the boundary element software like PROCAT and BEASY, and the finite element software like Elsyca Corrosion Master and COMSOL, etc., while the numerical simulation software related to cathodic protection has also been developed by Beijing University of Science and Technology. The accuracy of the corrosion prediction results is closely related to the boundary conditions of the numerical model, which generally means the relationship between the potential and current density of the corrosion electrode system. Strommen et al. [85] gave three types of boundary conditions when calculating the cathodic protection of offshore platforms, namely constant current density, linear polarization curve, and nonlinear polarization curve. The nonlinear polarization curve undoubtedly increases the difficulty of calculation, but it is the most representative and more common. Therefore, Iwata et al. [86] proposed a piecewise linearization method to solve this problem which accepted and quoted by other researchers.

With the advancement of electrochemical theory and measurement technology of thin electrolyte film, the corrosion prediction of aviation structural materials has aroused a new research upsurge. In 2009, Peratta et al. [87] introduced the galvanic corrosion modeling of typical macrostructures in aircraft at the European Corrosion Congress. By this means, they proved that the experimentally measured potential distribution and total galvanic current are highly consistent with the calculation results of the boundary element. Shi et al. [88] modeled and evaluated the galvanic interaction between Al 7075 alloy and noble potential materials by targeting model geometry, noble potential material types, and solution composition as influences. It shows that the galvanic action greatly affected initiation and expansion of localize corrosion for aluminum alloys. Thebault et al. [89] used the finite element method to simulate the Cu-Zn bimetal corrosion under the thin electrolyte film. Taking into account the transfer of $O_2$ in the electrolyte, the model calculated current density in the electrode edge by the scanning vibrating electrode technique (SVET), which basically matches the calculated value. Mizuno et al. [90] simulated the galvanic corrosion behavior of Al 5083 alloy and AISI4340 steel in atmospheric environment, and predicted the inter-granular corrosion damage of Al 5083 alloy caused by galvanic interaction. Cross et al. [91] used a time-dependent finite element model to study galvanic corrosion between aluminum and zinc coatings on steel surfaces.

Although the corrosion prediction technology of Al 7XXX alloy used in aircraft structure is becoming more perfect, there are several questions remained to be solved, such as how to determine the corrosive environment on the surface of aircraft structures, how to accurately measure the electrochemical properties of materials in atmospheric environments and how to select the corrosion prediction model [92–96].

First, the determination of the corrosion environment on the surface of aircraft structures is the basis of the corrosion prediction. The aircraft structures are complex and the corrosion environment is changeable. The corrosion medium on the structure surface can be simply divided into solution and thin electrolyte film only according to different positions in many studies. It is necessary to carry out further research on the causes of thin electrolyte film and the influencing factors of electrolyte film thickness. In addition, the corrosive medium is existed in a large number of cracks in the aircraft structure, which forms an oxygen concentration difference cell. The effect of this factor on the electrolyte film thickness also needs to be lucubrated. For another, accurate and reliable electrochemical measurement data is the guarantee of the corrosion prediction. At present, solution electrochemical measurement technology is relatively mature. Nevertheless, there are two problems in the measurement of electrochemical performance of thin electrolyte film. On the one hand, the change of the electrolyte film state affects the electrode reaction mass transfer process. On the other hand, it is difficult to maintain stability of electrolyte film thickness so that it influences the accuracy in the measurement. Further systematic study is needed for the influence of marine atmospheric environmental factors, including $Cl^-$ concentration, temperature, and PH. Moreover, the selection of the appropriate corrosion prediction model is a vital process of the corrosion prediction. A steady-state corrosion field can be used to model for galvanic corrosion of typical macrostructures in solution or thin electrolyte film. Whereas the corrosion medium changes continuously with time for the corrosion that occurs in narrow cracks, so that it is no longer appropriate to model through the steady-state corrosion field. At present, there are few studies on transient prediction of cracking corrosion in aircraft structures, and further research is needed on the quantitative analysis of corrosion process and influencing factors.

## 4. Main Measures to Improve the Performance of Al 7XXX Alloys for Aircraft Structures

The purpose of heat treatment for Al 7XXX alloys is to optimize the three microstructure parameters of matrix precipitates (MPt), grain boundary precipitates (GBPs), and precipitate-free zone (PFZ), so that the alloys have good comprehensive properties. The heat treatment process of Al 7XXX alloys mainly includes homogenization, solid solution, quenching, and aging. The microstructure evolution of Al 7XXX alloys is shown in Figure 8 at different heat treatment stages. It can be seen that from the as-cast state to the uniform annealed state, the spherical intra-granular non-equilibrium eutectic phase and a network of the coarse non-equilibrium eutectic phase at the grain boundaries were basically re-dissolved into the aluminum matrix to form a supersaturated solid solution, with only a small amount of the impurity phases remained. From the uniform annealed state to the final state, a large number of tiny and diffuse needle-like and spherical particles are precipitated within the grains, and the grain boundary precipitates are distributed in chains along the grain boundaries [20,21,97,98]. We mainly discuss the effects of solid solution, quenching, and aging on the microstructure and properties of Al 7XXX alloys.

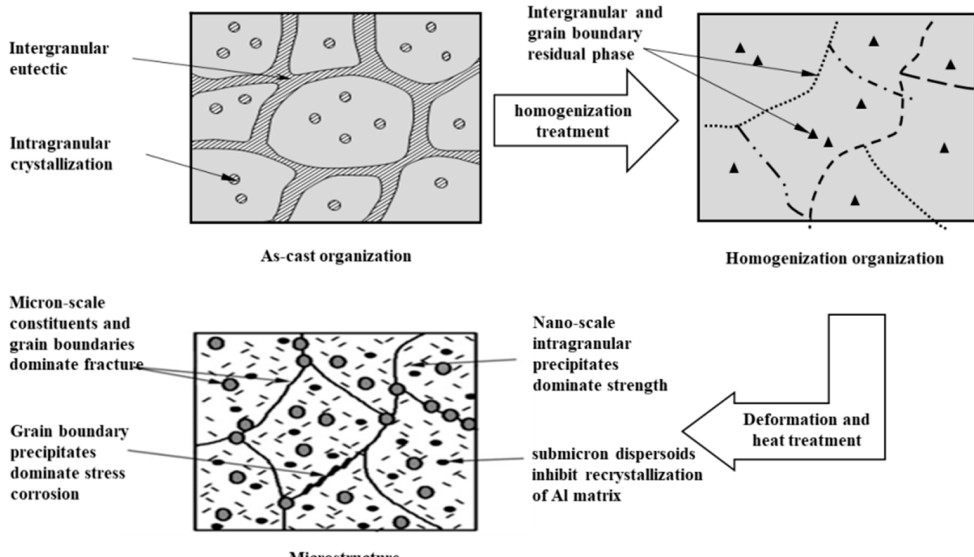

**Figure 8.** Microstructure evolution of Al 7XXX alloys [97,98].

*4.1. Solid Solution*

Solid solution is the basis of the heat-treated strengthening aluminum alloys to obtain high strength. It aims to fully dissolve the soluble elements in the alloy into the aluminum matrix to form a nearly uniformly distributed supersaturated solution, which facilitate subsequent aging precipitation to strengthen the alloy. The basic operations of the solution treatment are heating and holding. The solution temperature and holding time are the two most important parameters that determine the effect of solution. If the solution temperature is higher and the holding time is longer, the diffusion of solute atoms is the more favorable, so that the alloy elements are more fully dissolved and the aging effect is better. However, the recrystallization fraction will be increased by high temperature, and long-term holding which will adversely affect the strength, fracture toughness, and stress corrosion resistance after aging. Therefore, a good solid solution regime should be able to dissolve the soluble second phase as much as possible into the aluminum matrix without significantly increasing the recrystallization fraction in the alloy [99].

The solid solution regime for Al 7XXX alloys has developed from single-stage to multi-stage solid solution. Single-stage solid solution, as shown in Figure 9a, refers to enhancing the solid solution effect of the alloy element by simply increasing the final solid solution temperature and extending the solid solution time on the condition of avoiding over-burning. The disadvantage of this method is that the solid solution degree of alloying elements and the recrystallization fraction of the alloy increase simultaneously with the increase of solution temperature and time, and the comprehensive properties after ageing are poor [100]. Some scholars have proposed a stepwise temperature-increasing solution treatment to increase the solid solution degree of alloying elements while effectively suppressing the increase of the recrystallization fraction of the alloy. The stepwise temperature-increasing solid solution is shown in Figure 9b, which means that the alloy is first hold at a lower temperature for a certain time, and then gradually heated up to a higher temperature and held on. Through low-temperature solid solution, the low-melting non-equilibrium eutectic phase can be preferentially dissolved, and then gradually increased to exceed the multi-phase eutectic temperature, which promotes the maximum dissolution of the soluble second phase in the alloy. Simultaneously, the recovery takes place in the alloy during the low-temperature holding process, which suppresses the recrystallization in the subsequent high-temperature solid solution, so that the comprehensive properties of the alloy are significantly improved. In order to improve the stress corrosion resistance of Al 7XXX alloys, the near-solvus pre-precipitation following after high temperature solution treatment has been proposed, which means that solid solution occurs fully at high

temperature and then holds near the limit temperature of solid solution [101]. As shown in Figure 9c, when cooling from a high temperature to a lower temperature and holding, the super-saturation of the alloy is reduced with the temperature decreasing. When cooling at a slower rate from a high temperature to a lower temperature for holding, due to the low speed of cooling, the temperature gradient changes a little, so that the precipitation power is small, and the precipitates preferentially nucleates at the grain boundaries. In the subsequent aging process, the aging precipitates can grow up on the basis of the original precipitates. Thereby, the grain boundary precipitates becomes coarse and is discontinuous distribution, and the resistance to stress corrosion is improved. On the other hand, the intra-granular precipitates is tiny and dispersed, and the alloy has high strength [102].

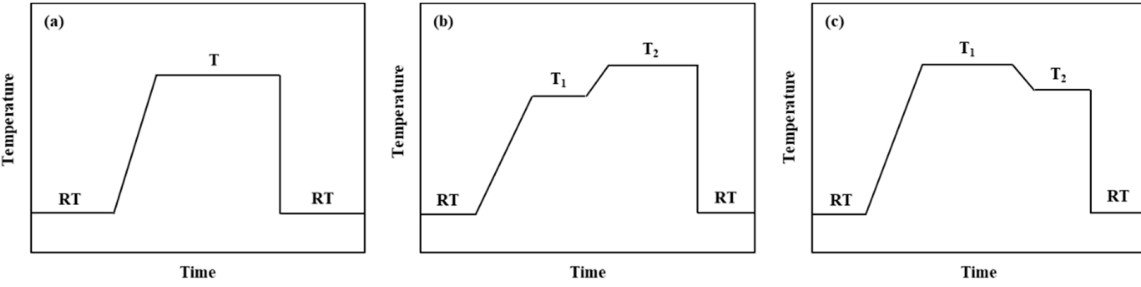

**Figure 9.** (**a**) Single-stage solid solution; (**b**) Stepwise temperature-increasing solution; (**c**) Near-solvus pre-precipitation following after high temperature solution [100–102].

During the solid solution process, the grain size, undissolved second phase fraction and solute atom distribution of the alloy will be changed, which affects the mechanical properties. Xu et al. [103] performed multi-stage solid solution on aluminum alloy extrusion materials and characterized and tested their microstructure and mechanical properties. The specific solid solution process is shown in Figure 10a. The results are shown in Figure 10b–d. With the solid solution temperature and time increased, its strength increases to the maximum at G3 and then decreases. In the solid solution process on the conditions of 450 °C × 2 h + 460 °C × 2 h + 470 °C × 2 h, and the aging process on 121 °C × 5 h + 133 °C × 16 h, aluminum alloy has excellent comprehensive properties with strength of 828.0 M Pa and elongation of 8.1%.

The study of Al 7XXX alloys found that the susceptibility to exfoliation corrosion and stress corrosion decreases first and then increases with the increase of solution temperature and time. It is mainly caused by the continuous decrease of the undissolved second phase and the constantly increase of recrystallization fraction. Shatry et al. [100] studied the effect of solution temperature on the stress corrosion properties of Al 7075 alloy. They found that the elements, such as Mg, Zn, Cu, and so on, migrated to the grain boundaries during the solution process. In addition with the increase of the solution temperature, degree of atomic aggregation at the grain boundaries decreases first and then increases, resulting in the susceptibility to stress corrosion decreases first and then increases.

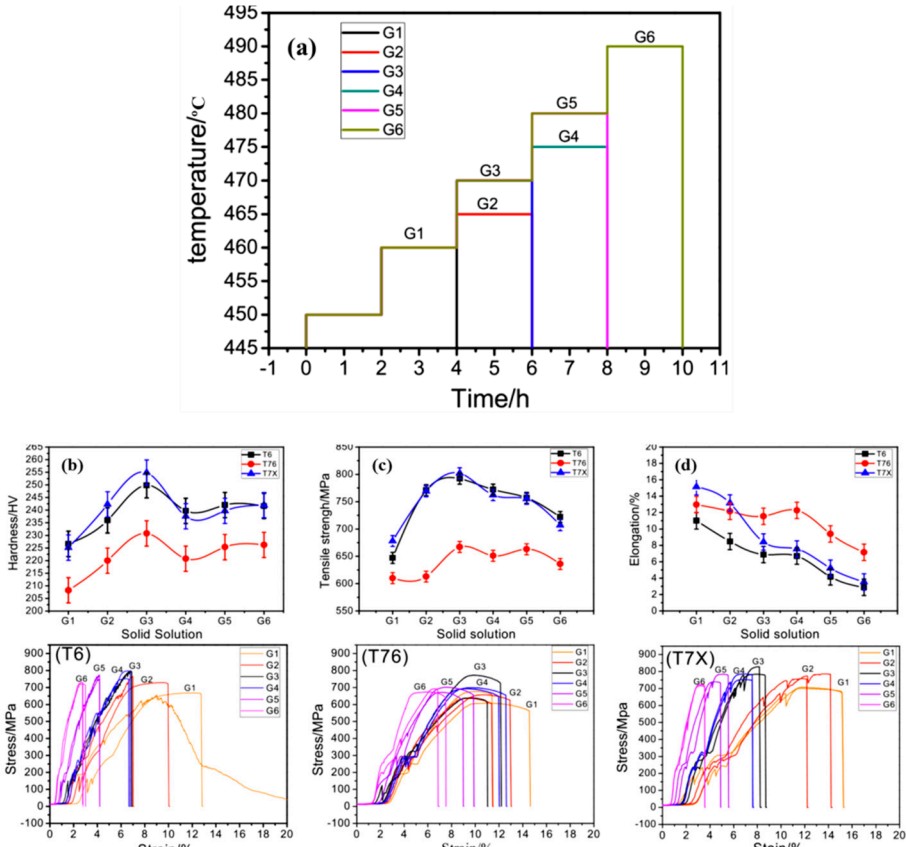

**Figure 10.** (**a**) Schematic diagram of solution process; Mechanical properties of alloys: (**b**) hardness, (**c**) tensile strength, (**d**) elongation and the corresponding stress-strain curves under T6, T73, T7X aging. (Reprinted with permission from ref. [103]. Copyright 2019 Elsevier).

*4.2. Quenching*

Quenching refers to the operation of rapidly cooling the aluminum alloy after solid solution to near room temperature through a certain medium, such as cold water, oil, etc. The intent of quenching is to fix the supersaturated solid solution in a fast cooling manner, so that the alloy maintains a certain solute super-saturation and vacancy concentration, so as to facilitate the diffusion of atoms and the formation of strengthening phases in the subsequent aging process [104].

Quenching transfer time and cooling rate are two crucial parameters that affect the properties of the age treated alloy. When quenching, the Al 7XXX alloys need to be transferred from the solid solution equipment to the quenching equipment [105]. As the alloy temperature continues to decrease, the solute atom super-saturation continuously increases, and the second phase is easy to be precipitated from the supersaturated solid solution [106]. However, the atomic diffusion rate will constantly decrease as the temperature reducing, resulting in difficulties in the precipitation of the second phase. When the alloy is lowered to a certain intermediate temperature, the solute super-saturation and the atomic diffusion rate are relatively high, and the second phase precipitation rate comes up to the maximum. The intermediate temperature range is referred to as the quenching sensitive range of the alloy. When the quenching transfer time is long or the quenching cooling rate is slow, the η phase is easily precipitated at the grain boundaries of the Al 7XXX alloys. After aging, η phase coverage at the grain boundaries of the alloy is high, which results in a high inter-granular corrosion sensitivity [107]. In addition, the large amount of precipitation of the η phase reduces the super-saturation of the alloy, so that the aging precipitation is difficult, and the strength of the alloy is reduced [108]. Therefore, in order to obtain high comprehensive properties of Al 7XXX alloy, the quenching transfer time must be strictly

controlled to avoid the alloy temperature falling to the quenching sensitive range, and the alloy temperature must be rapidly reduced below the quenching sensitive range with a fast cooling rate [27,107,109–113].

Song et al. [110] explored the effect of different quenching transfer time on the exfoliation corrosion of Al 7050-T6 alloy. As shown in Figure 11, the resistance to exfoliation corrosion of the alloy decreases with the increase of the quenching transfer time. As demonstrated in the figure, the coverage rate and the microstructure of the grain boundary precipitates are the most important factors affecting the properties of the alloy, and the solute depleted region at the grain boundaries has a small even no effect on the properties.

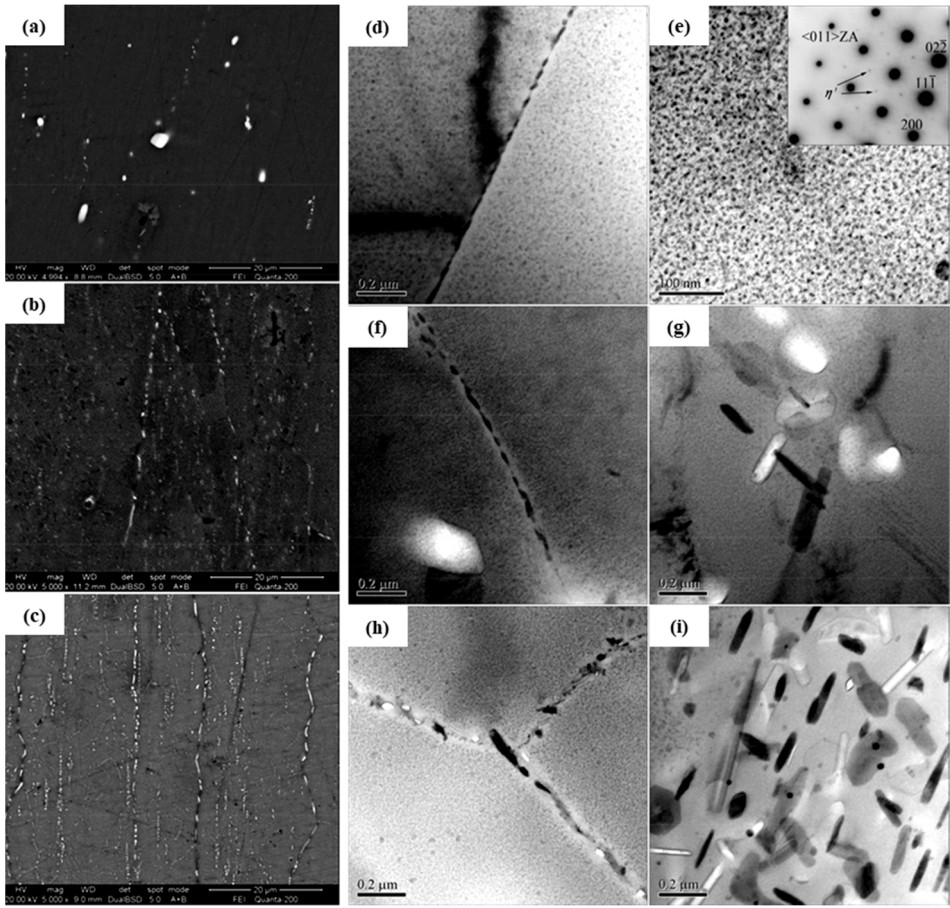

**Figure 11.** SEM images showing grain and sub-grain boundary precipitates in 7050-T6 alloys treated with different transfer times (**a**) 2 s, (**b**) 45 s, (**c**) 120 s; TEM images of 7050-T6 alloys treated with different transfer time (**d**,**e**) 2 s, (**f**,**g**) 45 s, (**h**,**i**) 120 s. (Reprinted with permission from ref. [110]. Copyright 2014 Elsevier).

The chemical composition is the most important factor affecting the quenching sensitivity of the alloy. The quenching sensitivity will be changed with the variation for the content or proportion of Zn, Mg, and Cu elements in the alloy. Yuan et al. [114] studied the corrosion behavior of Al-Zn-Mg-Cu alloys at different Cu contents and quenching rates. As shown in Figure 12, through the analysis of the tensile strength and elongation of the alloy, it was found that with the same Cu content, as the quenching rate decreases, the tensile strength decreases, and the elongation increases first and then decreases. Slow quenching rate can effectively improve the SCC resistance of Cu-low alloys, especially reducing the crack growth rate, which means that SCC propagation velocity of Cu-low alloys with the slow-quenching-rate and is about an order of magnitude lower than that with the fast quenching rate. However, the slow quenching rate has less effect on the Cu-rich alloys.

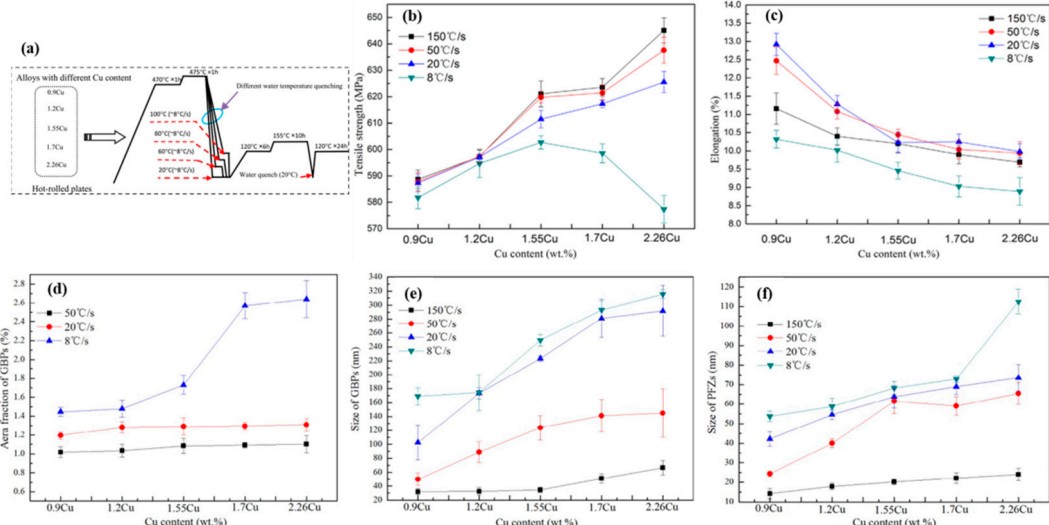

**Figure 12.** (**a**) Schematic diagram of heat treatment processing for alloys; (**b**,**c**) Tensile strength and elongation curve of the well-quenched and aged samples treated by different quench rate; (**d**) Effect of Cu content and quench rate on area fraction of GBPs in alloys; (**e**,**f**) Effect of Cu content and quench rate on the size of GBPs and width of PFZs in alloys. (Reprinted from ref. [114]).

During quenching, due to the different cooling rates of the surface and the core in the Al 7XXX alloys, the microstructure and residual stress of the alloys are unevenly distributed, resulting in reduction in the stress corrosion resistance. Therefore, a new quenching process is required to improve the alloy properties. Xie et al. [115] studied the effect of step quenching and aging heat treatment on the stress corrosion cracking properties of the alloys. The results show that step quenching can significantly improve the stress corrosion resistance. As shown in Figure 13, the stress corrosion cracking resistance has been significantly improved through step quenching and aging heat treatment, but it has been hardly improved through the regression aging and two-stage over-aging treatment compared with peak aging. For Cu-low Al-Zn-Mg-Cu alloys, the stress corrosion resistance was improved after step quenching and aging heat treatment, mainly because of the high Cu content, large size, and discontinuous distribution of grain boundary precipitates.

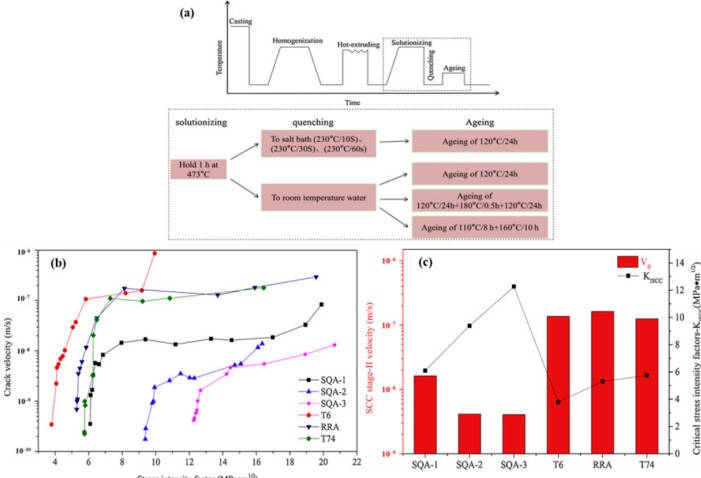

**Figure 13.** (**a**) Schematic diagram of heat treatment for the alloys investigated; (**b**) dependence of stress corrosion cracking propagation rate (v) and stress intensity factor ($K_I$) of different heat treatments specimens; (**c**) $V_{II}$ and $K_{ISCC}$ values of different heat treatments. (Reprinted with permission from ref. [115]. Copyright 2019 Elsevier).

Chen et al. [27] explored the effect of quenching rate on the microstructure and stress corrosion properties of Al 7085 alloy. With the decrease of the quenching rate, the size and inter-particle distance of the grain boundary precipitates and precipitation free zone width increase, but the Cu content in the precipitates decreases. In addition, the stress corrosion resistance of the alloy increases first and then decreases with the decrease of the quenching rate, as shown in Figure 14. According to analysis, the size, distribution, and the Cu content of the precipitations at the grain boundaries are the main factors affecting the stress corrosion resistance of the alloy.

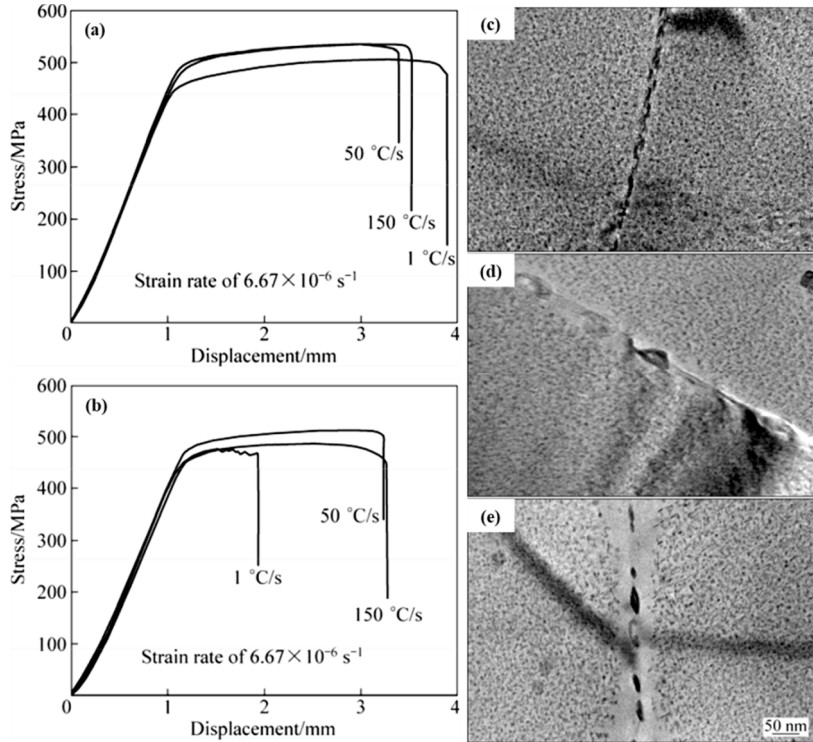

**Figure 14.** SSRT results for AA7085 with different quenching rates (**a**) in air (**b**) in 3% NaCl + 0.5% $H_2O_2$ solution; TEM microstructures of alloys with different quenching conditions: (**c**) 150 °C/s, (**d**) 50 °C/s, (**e**) 1 °C/s. (Reprinted with permission from ref. [27]. Copyright 2012 Elsevier).

*4.3. Aging*

Aging is a main method to optimize the microstructure and comprehensive properties of Al 7XXX alloys. After solid solution and quenching of Al 7XXX alloys, the alloy elements are in a supersaturated state, and the dislocation density is high. While kept at room temperature, that is natural aging, it is easy for the second phase, which means mainly GP zones in Al 7XXX alloys, to precipitate from the alloy, so that the alloy is strengthened [80,116]. However, this process is extremely long. Even after kept for a year, the alloy cannot reach a stable state, and its strength is still slowly rising [117], as shown in Figure 15.

In the actual production process, artificial aging is usually used to achieve the desired properties of alloys. That is, the aging precipitation process of the quenched alloy is accelerated by holding at a higher temperature [118]. The artificial aging state of Al 7XXX alloys has in turn experienced the development process of peak aging (T6), over-aging (T7X), and retrogression and re-aging (RRA). In general, the precipitation order of the second phase in the Al 7XXX alloys during the artificial aging process is supersaturated solid solution (SSS), to vacancy rich clusters, to GPIIzones, to η′ (spherical), and to η (platelet) [119–121]. In Al 7XXX alloys for aircraft, despite the second phases with GP zones → η′ → η ($MgZn_2$), other precipitates can be also formed [42,119,122] as shown in Table 4.

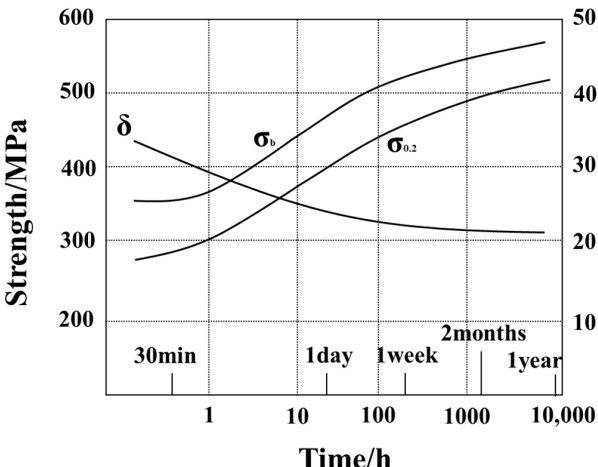

**Figure 15.** Mechanical properties of the Al 7XXX alloys during natural aging [116].

**Table 4.** Observed precipitates in aircraft Al alloy products [42,119,122].

| Alloy | Observed Precipitate |
|-------|----------------------|
| 7X75 | $Al_{12}Mg_2Cr$ |
| 7X50 | $Al_3Zr$ |
| 7055 | $Al_3Zr$ |
| 7X75-T6 | h' (a precursor to h, $MgZn_2$ or $Mg(Zn,Cu,Al)_2$) |

### 4.3.1. Peak Aging

Peak aging is an aging heat treatment method that maximizes the strength of the alloy. In order to make the alloy obtain the peak strength, it is necessary to precipitate tiny and dispersed particles within the grains to obstruct the dislocation motion during deformation. Only the coherent GP zones and the semi-coherent η'($MgZn_2$) phase with the matrix in Al 7XXX alloys can effectively pin mobile dislocations and are functioned as strengthening [123]. Therefore, the aging temperature should be controlled below the melting point of the GP zones, or the heating rate should be controlled, so that a large number of GP zones are first precipitated within the grains. Then it is appropriately transformed to η' phase. Finally, the tiny and dispersed particles mainly including the GP zones and η' phase are formed in the grains, which makes the alloy obtain the highest strength [41]. However, because the aging temperature is often low or the aging time is relatively short, the alloy elements, especially Cu element, occur incomplete diffusion, resulting in a high potential difference between the grains and grain boundaries in this heat treatment method. Meanwhile, as shown in Figure 16, due to the continuity of chain-like grain boundary phases of the T6 state alloy, it is easy to become a continuous channel for corrosion expansion, so that the localized corrosion susceptibility is high.

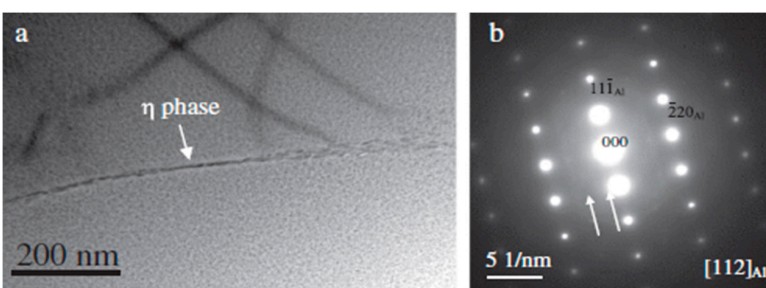

**Figure 16.** Distribution of grain boundary precipitates of T6 high strength aluminum alloy (**a**) TEM (**b**) SADPs. (Reprinted with permission from ref. [120]. Copyright 2015 Elsevier).

4.3.2. Over-Aging

An over-aging heat treatment method is developed to improve the corrosion resistance of the T6 state alloy. After over-aging heat treatment, the corrosion resistance of the alloy is improved, and the residual stress is correspondingly reduced. Also, the dimensional stability of the alloy in high temperature environment is enhanced, which makes the service environment and service life of the alloy expanded and improved. However, this comes at the expense of losing partial strength. The strength of the Al-Zn-Mg-Cu alloy in the T73 and T74 states is 10–15% lower than that of the T6 state alloy according to the literature [124,125].

In order to shorten the time for the alloy to reach the over-ageing state, the over-aging treatment usually adopts two-stage aging, including the aging heat treatment method of low temperature first, then high temperature or high temperature first and then low temperature. T74 (T736) is a typical over-ageing heat treatment through low temperature treatment followed by high temperature treatment. Its first-stage low-temperature aging is pre-aging with the purpose of precipitating fine and dispersed GP zones in the alloy grains. The second-stage high-temperature aging is a stabilization stage. During the aging process, the GP zones gradually transforms to η′ and η phase and grows, and the grain boundary precipitates are coarse and discontinuously distributed. Also, the intra-granular precipitates grow and are distributed unevenly. Ultimately, the corrosion resistance of the alloy increases, and the strength decreases, as shown in Figure 17a [126–128]. The high-temperature followed by low-temperature aging heat treatment method is to re-dissolve a small amount of GP zones through high-temperature aging, which increase the degree of super-saturation of the alloy, and then precipitate a large number of fine and dispersed strengthening phases within grains through low-temperature aging. The partial η phase can be precipitated during the high-temperature aging stage. Due to the high melting point of the η phase, it is difficult to dissolve at the high-temperature stage. It can become a nucleation point at the low-temperature aging stage, which promotes the nucleation and growth of the grain boundary precipitates. As a result, the precipitates are coarse and discontinuously distributed, as shown in Figure 17b [128].

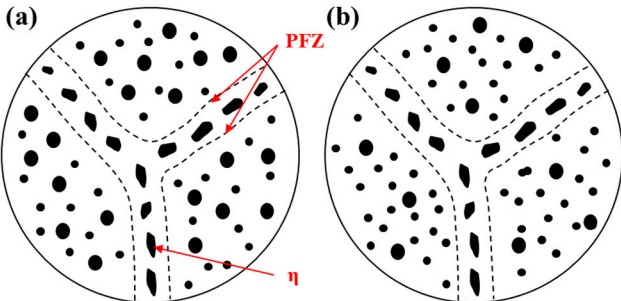

**Figure 17.** Precipitate distribution near the grain boundary of two-stage over aging (**a**) T74; (**b**) high-temperature followed by low-temperature aging [126–128].

4.3.3. Retrogression and Re-Aging

In order to make Al 7XXX alloys have high corrosion resistance while maintaining high strength, Cina et al. [129] proposed the heat treatment process of retrogression and re-aging(RRA) in 1974. Retrogression and re-aging is actually a three-stage aging treatment method, which includes pre-aging, retrogression, and re-aging. Pre-aging means to keep at a low temperature for a long period of time. Retrogression is always through a comparatively high temperature for a short time. Re-aging refers to keep at a lower temperature for a long period [130].

After RRA treatment, the strength of the alloy is close to the peak, and the stress corrosion resistance, fracture toughness, and fatigue resistance are significantly improved [131–134]. It is attributed to the regulation of RRA on the microstructure of the alloy, including the

composition, morphology, and distribution of the precipitates. However, there are different views which were put forward on the microstructure evolutions at different stages based on the research results. In general, for the RRA treatment, the GP zones are precipitated during the pre-aging stage and properly transforms to the η′ phase and grows. In the stage of high-temperature retrogression, the finer GP zones and η′ phase are re-dissolved, while the grain boundary precipitates have almost no re-dissolution due to the large size and that has been transformed into stable η phase [37,131]. During the re-aging process, the fine and dispersed GP zones and η′ phase are re-precipitated in the alloy grains, and the grain boundary precipitates can nucleate and grow based on the existing precipitates, which eventually become discontinuously distributed [36,135–137], as shown in Figure 18.

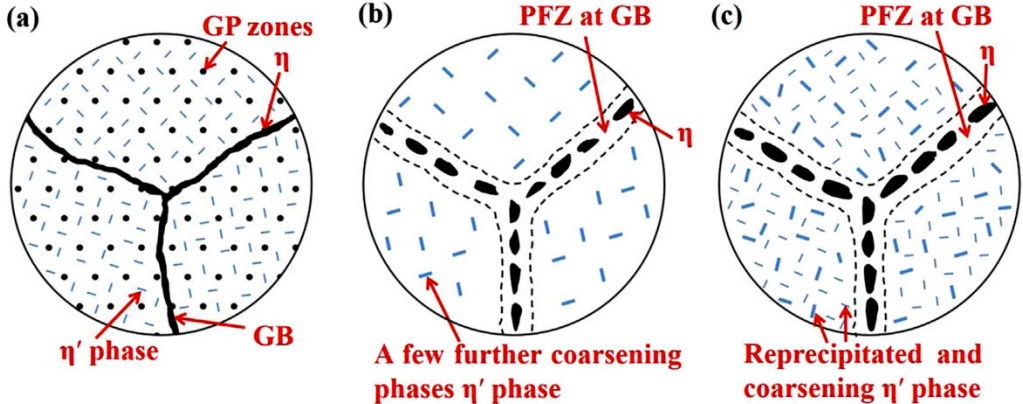

**Figure 18.** Microstructure changes near grain boundaries during RRA treatment: (**a**) pre-aging (**b**) retrogression (**c**) re-aging. (Reprinted with permission from ref. [36]. Copyright 2020 Elsevier).

Yang et al. [120] studied the mechanical and corrosion resistance properties of Al-6.0Zn-2.3Mg-1.8Cu-0.1Zr (wt %) alloy on different aging treatment conditions. As shown in Figure 19, the RRA state has higher strength and better corrosion resistance than the routine T6 and T74 states. The optimal process for the experimental alloy is on the conditions of 120 °C × 24 h + 180 °C × 60 min + 120 °C × 24 h.

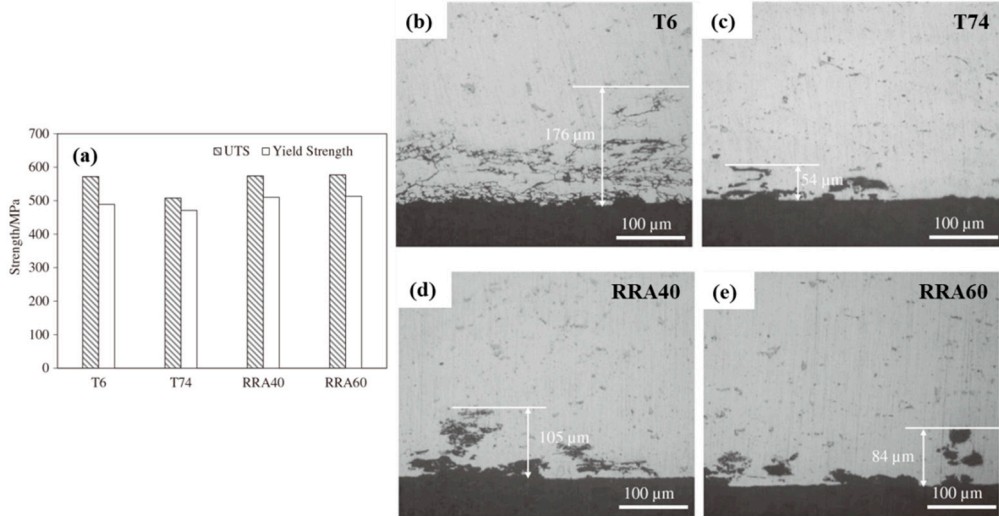

**Figure 19.** (**a**) Mechanical properties; Intergranular corrosion morphology of the alloy under different aging treatment conditions: (**b**) T6, (**c**) T74, (**d**) RRA40, (**e**) RRA60. (Reprinted with permission from ref. [120]. Copyright 2015 Elsevier).

Chen et al. [41] explored the effects of aging treatment on stress corrosion cracking, fracture toughness, and strength of Al 7085 alloy. As shown in Figure 20, compared with

T6 state, the fracture toughness of the T74 state is improved by 22.9%, but the strength is reduced by 13.6%. The fracture toughness of the RRA state is 14.2% higher than that of the T6 state, and the strength is almost unchanged. The fracture toughness of the DRRA and T74 state is comparable, but the former strength is increased by 14.6% which compared to the latter.

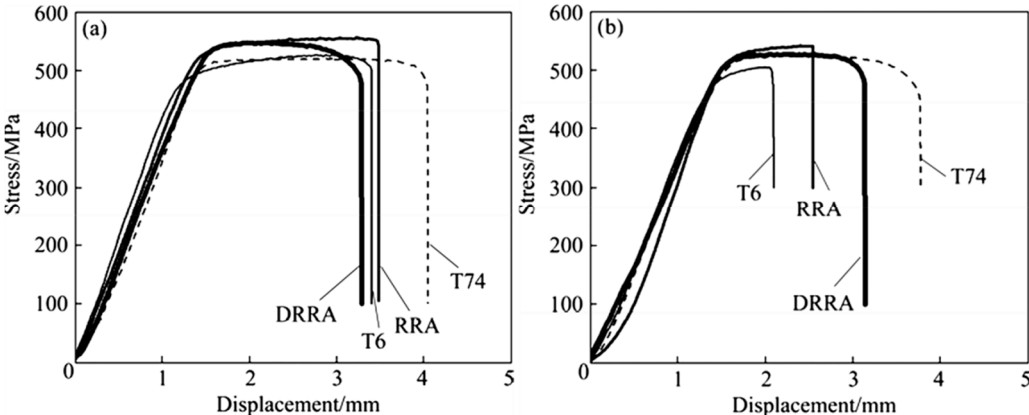

**Figure 20.** SSRT results of AA7085 under different aging regimes (**a**) in air (**b**) in 3% NaCl solution. (Reprinted with permission from ref. [41]. Copyright 2014 Elsevier).

Lin et al. [138] researched the effects of different aging treatments on the tensile strength and stress corrosion resistance of Al 7050 alloy in a 3.5% NaCl solution at pH12. As shown in Figure 21, the tensile strength is improved in T6 state, but the stress corrosion resistance is reduced. In contrast, the stress corrosion resistance of T74 state is improved, but the tensile strength is reduced. The tensile strength and stress corrosion resistance are both enhanced in the RRA state which is much better than the other two methods.

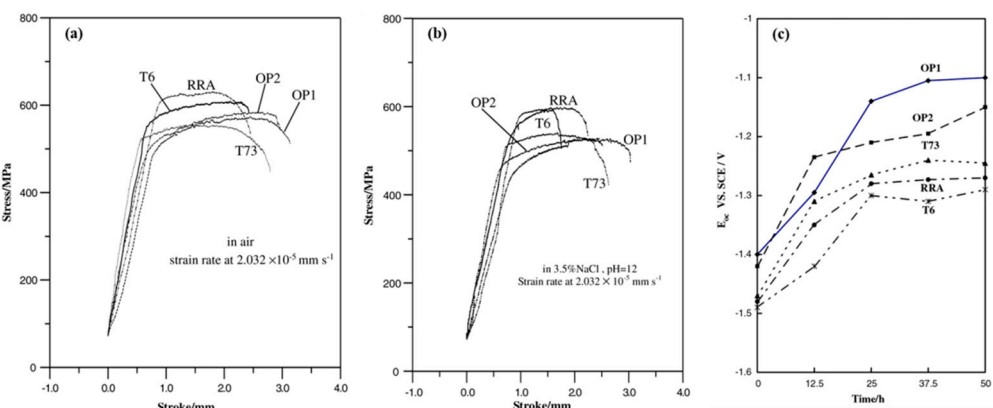

**Figure 21.** SSRT results of 7050 with different aging regimes (**a**) in air (**b**) in 3% NaCl solution; (**c**) Variation of corrosion potential with time for various specimens in 3.5% NaCl solution at pH12. (Reprinted with permission from ref. [138]. Copyright 2006 Elsevier).

Wang et al. [36] explored the effect of RRA on the microstructure, hardness and corrosion resistance of Al 7085 alloy. It shows that the properties of Al 7085 alloy are sensitive to tempering temperature and time. As shown in Figure 22, the alloy can obtain good mechanical and corrosion resistance properties on condition of 120 °C × 24 h + 160 °C × 1.5 h + 120 °C × 24 h. Compared with the peak aging, the hardness of RRA is increased by 10.2%, which mainly attribute to the dispersed rod-like η phase in the matrix. Also, the corrosion resistance is improved due to the coarse η′ phase of the discrete distribution formed by the higher Cu content and the narrow precipitate-free zone which are approximately 45–50 nm along the grain boundaries.

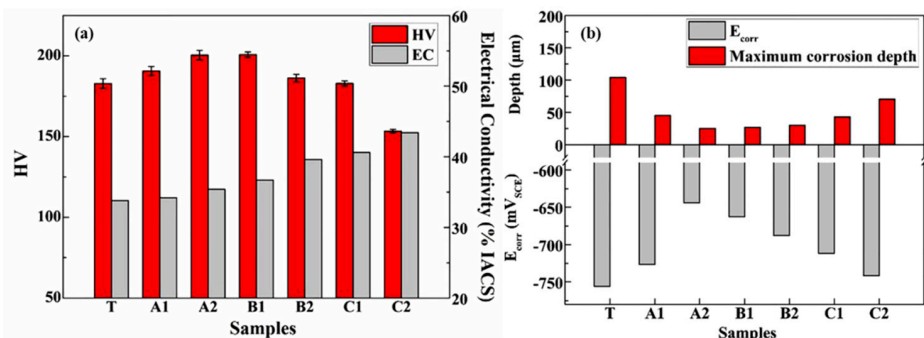

**Figure 22.** (**a**) Hardness and conductivity of 7085 alloy with different aging conditions; (**b**) Effect of retrogression treatment on the values of Ecorr and maximum corrosion depth in IGC test. (Reprinted with permission from ref. [36]. Copyright 2020 Elsevier).

## 5. Conclusions

As the main part of high strength aluminum alloy, Al 7XXX alloys have been successfully used as the main materials of aircraft structural components. With the application of titanium alloys and composite materials in the fuselage design, the proportion of aluminum alloy has been reduced. In order for aluminum alloys to remain attractive in the airframe construction, research should be necessarily carried out in terms of structural properties, weight reduction, and cost reduction. Therefore, current studies for Al 7XXX alloys contain improvements on mechanical properties; reduction of manufacturing, maintenance, and repair costs; prevention of corrosion and fatigue; and ability to perform reliably throughout its service life.

In recent years, Al 7XXX alloys have successfully improved static strength, fracture toughness, fatigue and corrosion resistance through composition design and control of chemical composition, as well as via the exploitation of more efficient heat treatment methods. It can be seen from this review that the main improvement of Al 7XXX alloys is to optimize the solute content and solute ratio to achieve better balance for the performances. Therefore, for the design of the alloy, the content of Zn will be increased to more than 10%, while the content of Mg and Cu will be reduced. Also, the content of impurity elements such as Fe and Si will be even lower. On the other hand, the addition of trace transition elements like Zr and Er will be more reasonable. Accordingly, $MgZn_2$ phase is the main strengthening precipitate in Al 7XXX alloys. The formation, distribution and geometrical specifications of $MgZn_2$ phase are highly sensitive to the processing parameters of ageing and the way it proceeds. In order to better manipulate the microstructure and obtain the best mechanical and corrosion properties, various aging processes have been developed for Al 7XXX alloys. At present, the heat treatment regime is developed along T6, T73, T76, T736 (T74), to T77 (T78, T79). In order to obtain better comprehensive properties of Al 7XXX alloys, it is necessary to improve the existing heat treatment regime or develop a new one. In brief, the development of new generation Al 7XXX alloys for aircraft structure should give consideration to high strength, high toughness, high damage tolerance, high quenching, and good corrosion resistance.

The developments of manufacturing techniques are the key issues for the weight and cost reduction except for the improvement on the structural performance. Manufacturing occupies the biggest portion of the fuselage cost. Therefore, novel assembly techniques including laser beam welding and friction-stir welding, high-speed machining and AM technique should be introduced to reduce the production costs and part count.

In addition, Al 7XXX alloys is susceptible to corrosion by environment during aircraft service, which has an impact on the life and reliability of aircraft. Therefore, corrosion prediction technology has a strong practical significance for flight safety. In order to solve the problem of corrosion prediction of Al 7XXX alloys for aircraft structures, the surface corrosion environment needs to be confirmed first, and the accurate and reliable electrochemical measurements are required. In this way, the corrosion behavior of Al

7XXX alloys for aircraft structures can be accurately predicted by constructing a reasonable prediction model.

**Author Contributions:** Conceptualization, B.Z. and S.Z.; literature search, B.Z.; figures, B.Z.; data collection, B.Z.; data analysis, B.Z.; data interpretation, B.Z.; writing—original draft preparation, B.Z.; writing—review and editing, B.Z. and S.Z.; supervision, B.L. and S.Z.; validation, B.L. and S.Z. All authors have read and agreed to the published version of the manuscript.

**Funding:** This research was funded by National key R & D projects, grant nos. 2019YFC1907101, 2019YFC1907103, 2017YFB0702304, Key R & D project in Ningxia Hui Autonomous Region, grant no. 2020BCE01001, key and normal projects National Natural Science Foundation of China, grant nos. U2002212, 51672024, Xijiang Innovation and Entrepreneurship Team, grant no. 2017A0109004, the Fundamental Research Funds for the Central Universities, grant nos. FRF-BD-20-24A, FRF-TP-20-031A1, FRF-IC-19-017Z, FRF-GF-19-032B, FRF-TP-20-097A1Z, 06500141, and Financial support from the State Key Laboratory for Advanced Metals and Materials, grant no. 2019Z-05.

**Data Availability Statement:** Not applicable.

**Acknowledgments:** The authors would like to thank the editor for editing of the manuscript and the anonymous reviewers for their detailed and helpful comments.

**Conflicts of Interest:** The authors declare no conflict of interest.

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
