# Peer review of "The Advancement of 7XXX Series Aluminum Alloys for Aircraft Structures: A Review"

_metals, doi:10.3390/met11050718_

Round 1

Reviewer 1 Report

The review entitled: The advancement of 7XXX series aluminum alloys for aircraft structures: A literature review has been nicely articulated. I have the following comments/suggestions:

  • The motivation for presenting the present review should be highlighted in Section 1. Introduction - which is not the case in the present form.
  • Fig. 1 and 2 are squeezed and are not of publishable quality.
  • The fabrication of 7XXX series by additive manufacturing should be introduced and the issues involved in fabricated should also be discussed, otherwise, the present review is not complete.
  • In several of the figures, the references and copyright information is missing.
  • The English language needs attention.
  • The typos in the manuscript should be carefully rectified. For instance, the units and symbols should have space in between. Ex. - 179MPa should be written as 179 MPa.

Reviewer 2 Report

Please, see the file attached below.

Reviewer 3 Report

Please do not waste the opportunity to make paper better

Round 2

Reviewer 1 Report

The authors have reasonably addressed the comments and the manuscript may now be accepted for publication.

Author Response

Thanks very much for your professional and rapid processing of our paper.

Reviewer 2 Report

Accepted in the current form.

Author Response

Many thanks for the valuable comments and suggestions from the Reviewer.

Reviewer 3 Report

Paper is Ok

Author Response

We greatly appreciate the efficient and professional processing of our paper.